# Water-Soluble Polyoxometal Clusters of Molybdenum (V) with Pyrazole and Triazole: Synthesis and Study of Cytotoxicity and Antiviral Activity

**DOI:** 10.3390/molecules28248079

**Published:** 2023-12-14

**Authors:** Anna V. Konkova, Iulia V. Savina, Darya V. Evtushok, Tatiana N. Pozmogova, Maria V. Solomatina, Alina R. Nokhova, Alexander Y. Alekseev, Natalia V. Kuratieva, Ilia V. Eltsov, Vadim V. Yanshole, Aleksander M. Shestopalov, Anton A. Ivanov, Michael A. Shestopalov

**Affiliations:** 1Nikolaev Institute of Inorganic Chemistry SB RAS, 3 Acad. Lavrentiev Ave., Novosibirsk 630090, Russia; konkova@niic.nsc.ru (A.V.K.); savina@niic.nsc.ru (I.V.S.); evtushok@niic.nsc.ru (D.V.E.); pozmogova@niic.nsc.ru (T.N.P.); kuratieva@gmail.com (N.V.K.); shtopy@niic.nsc.ru (M.A.S.); 2Research Institute of Virology, Federal Research Center of Fundamental and Translational Medicine, 2 Timakova St, Novosibirsk 630117, Russia; mariaza@ngs.ru (M.V.S.); alina-nohova@mail.ru (A.R.N.); al-alexok@yandex.ru (A.Y.A.); shestopalov2@mail.ru (A.M.S.); 3Research Institute of Applied Ecology, Dagestan State University, 43a Gadzhiyeva St, Makhachkala 367000, Russia; 4Department of Natural Sciences, Novosibirsk State University, 2 Pirogova Str., Novosibirsk 630090, Russia; eiv@fen.nsu.ru; 5International Tomography Center SB RAS, 3a Institutskaya Str., Novosibirsk 630090, Russia; vadim.yanshole@tomo.nsc.ru; 6Department of Physics, Novosibirsk State University, 1 Pirogova St., Novosibirsk 630090, Russia

**Keywords:** polyoxometal cluster, molybdenum, pyrazole, triazole, cytotoxicity, antiviral activity

## Abstract

Among well-studied and actively developing compounds are polyoxometalates (POMs), which show application in many fields. Extending this class of compounds, we introduce a new subclass of polyoxometal clusters (POMCs) [Mo_12_O_28_(μ-L)_8_]^4−^ (L = pyrazolate (pz) or triazolate (1,2,3-trz or 1,2,4-trz)), structurally similar to POM, but containing binuclear Mo_2_O_4_ clusters linked by bridging oxo- and organic ligands. The complexes obtained by ampoule synthesis from the binuclear cluster [Mo_2_O_4_(C_2_O_4_)_2_(H_2_O)_2_]^2−^ in a melt of an organic ligand are soluble and stable in aqueous solutions. In addition to the detailed characterization in solid state and in aqueous solution, the biological properties of the compounds on normal and cancer cells were investigated, and antiviral activity against influenza A virus (subtype H5N1) was demonstrated.

## 1. Introduction

Polyoxometalates (POMs)—polyoxo compounds of transition metals in high oxidation states—form a large class of compounds with a wide variety of structures [1,2]. Availability, ease of synthesis and the possibility of chemical modification have made POMs promising objects for many studies. Depending on the structure of the POM, one can distinguish: (i) heteropolyanions (having heteroatoms in addition to metal atoms in the structure), (ii) isopolyanions (built only on metal oxide), and (iii) Mo-blue and -brown (having reduced metal centers in the structure) [1,3]. In addition to “classical” POMs (such as Lindqvist (Figure 1a), Wells–Dawson, Anderson, etc.-type POMs) [1], hybrid POMs (covalently bonded to organic ligands) [4], POM-based coordination polymers (POMOFs) [5,6,7], various supramolecular assemblies [8,9,10], etc., have been reported in the literature. This diversity of compounds has led to the identification of a number of potentially practical applications for the compounds, mainly in the fields of catalysis [11,12] or biology and medicine [13,14,15,16,17]. In addition to the compounds described above, there are complexes in the literature that are structurally similar to POM, but are based on bi- or tri-nuclear molybdenum oxo clusters (Mo^V^_2_O_4_ and Mo^IV^_3_O_4_) connected by oxo ligands, organic linkers, or by Mo^VI^O_6_-fragements. There are very few examples of compounds, and most of the work dates back to 1993–2004 (Table 1 and Figure 1b).

As one can see, there is no systematicity in these works and the yields of compounds are not high, except for a few complexes. Moreover, without taking into account more recent work, the compounds have often been studied only structurally, while their behavior in solution and any physicochemical properties have not been reported. There are a number of works in the literature devoted to POMOFs, such as (TBA)_3_[PMo^V^_8_Mo^VI^_4_O_36_(OH)_4_Zn_4_][C_6_H_3_(COO)_3_]_4/3_ (Figure 1c), which contain ε-Keggin POM, also based on binuclear clusters [5]. Such POMOFs, similar in structure to the compounds described above, have been shown to be effective electrocatalysts for the formation of hydrogen from water (hydrogen evolution reaction). Thus, a more detailed study of POM based on Mo-clusters may reveal their possible practical applications and produce new functional materials.

Aiming at a detailed study of such compounds, we focused our research on the possibility of obtaining new POMs with heterocycles — pyrazoles and triazoles. To increase the chances of POM formation based on dinuclear clusters, (NH_4_)_2_[Mo_2_O_4_(C_2_O_4_)_2_(H_2_O)_2_]·3H_2_O, which already contains a preformed dinuclear Mo_2_O_4_ cluster, was chosen as a starting compound. In this work, a series of new compounds were obtained, described by the general formula (NH_4_)_4_[Mo_12_O_28_(μ-L)_8_] (L = pyrazolate (pz) or triazolate (1,2,3-trz or 1,2,4-trz)) (Figure 1d). The synthesized compounds can be classified as both cluster compounds (having an Mo–Mo covalent bond in the structure) and hybrid POMs. By classifying these compounds into a separate class, in this work, we introduce the concept of polyoxometal cluster (POMC), which reflects the association of the obtained complexes with the chemistry of cluster compounds and POM. Mo_12_-POMC have been studied in detail both in solid state (single-crystal or powder X-ray diffraction and elemental analysis) and in aqueous solution (NMR spectroscopy and mass spectrometry). The solubility and stability in aqueous solutions at neutral and alkaline pH allowed first studies on the cytotoxicity of the compounds on normal and cancer cell cultures and antiviral activity against influenza A virus.

## 2. Results and Discussion

### 2.1. Synthesis of POMC

In contrast to the methods described in the literature for the preparation of various POMCs [18,19,20,21,22,23,24,25], in this work, the compounds were obtained by ampoule synthesis at high temperatures. The starting compound chosen was the complex (NH_4_)_2_[Mo_2_O_4_(C_2_O_4_)_2_(H_2_O)_2_]·3H_2_O, which already contains a preformed dinuclear Mo_2_O_4_ cluster. The synthesis was carried out in sealed glass ampoules in an excess of organic ligand (pyrazole, 1,2,3- or 1,2,4-triazoles) at 200 °C for 2 days. Under these conditions, the organic compound acts both as a ligand and as a reaction medium. During the reaction, dinuclear clusters self-assemble into POMC, the structure of which will be discussed below. Also at this temperature, the oxalic acid and ammonium oxalate formed during the synthesis decompose into NH_3_, CO_2_ and H_2_O, creating a high pressure in the ampoules, which may lead to an explosion of the ampoule (it is recommended to use ampoules with a volume of ∽8 cm^3^ per 200 mg of the initial complex to avoid an explosion of the ampoule). The reaction time and temperature are optimal: longer times do not increase the yield, while shorter times significantly reduce the yield of the compounds; higher temperatures lead to the decomposition of complexes, and lower temperatures lead to a mixture of products of unknown composition and structure. Upon slow cooling of the reaction mixture, crystals of the compounds (NH_4_)_4_[Mo_12_O_28_(μ-pz)_8_]·1.5pzH·4.5H_2_O (**1**), (NH_4_)_4_[Mo_12_O_28_(μ-1,2,4-trz)_8_]·1.5(1,2,4-trzH)·3H_2_O (**2**) and (NH_4_)_4_[Mo_12_O_28_(μ-1,2,3-trz)_8_]·4H_2_O (**3**) suitable for single crystal X-ray diffraction (SCXRD) analysis are obtained. Crystalline products in high yield (>90%) are easily separated from the reaction mixture by washing with diethyl ether, ethanol and a small amount of cold water. The composition and structure of the compounds have been confirmed by a complex of physicochemical analytical methods (see Section 3), including SCXRD, powder X-ray diffraction analysis, elemental analysis, IR spectroscopy, NMR spectroscopy and high-resolution electrospray mass spectrometry (HR-ESI-MS). The complexes are highly soluble in water, dimethyl sulfoxide and N,N-dimethylformamide, but stable only in aqueous solutions.

### 2.2. Crystal Structure

The crystal structure of all the compounds obtained was determined by single-crystal X-ray diffraction analysis (SCXRD). In all cases, single crystals suitable for SCXRD were obtained by slow cooling of the reaction mixture. According to the SCXRD, all structures contain a similar “barrel”-type motif—[Mo_12_O_28_L_8_]^4−^ (L = μ-pz, μ-1,2,3-trz or μ-1,2,4-trz)—with dimensions slightly smaller than the well-known Dawson-type POM [P_2_W_18_O_62_]^6−^ (Figure 2). These compounds are formed by binuclear clusters Mo_2_O_4_ linked by organic ligands and bridging μ-O- and μ_3_-O-ligands. In the structures, two types of binuclear clusters with different ligand environments can be distinguished, located at the top and bottom of the “barrel” or on the walls of the “barrel” (Figure 2). The first type (I) contains a Mo_2_O_4_ (O=Mo-(μ-O)^i^_2_-Mo=O, i = inner) cluster (with one Mo-Mo bond) coordinated by four organic ligands (two per Mo) and two oxo ligands (one per Mo, denoted as terminal). The second type (II) also contains a Mo_2_O_4_ cluster, but coordinated by four oxo ligands (two per Mo, denoted as terminal) and two organic ligands (one per Mo). The organic ligands are bridging and bind one type I and one type II cluster each. Terminal oxo ligands of type I clusters are μ_3_ ligands and bind one type I cluster and two type II clusters (terminal to both clusters). The μ-O^i^ ligands of the type I cluster Mo_2_O_4_ are also terminal ligands for type II clusters (μ_3_ type, binding two Mo in the type I cluster and one Mo from the type II cluster). One of the μ-O^i^ ligands type II cluster Mo_2_O_4_ is also the terminal ligand for the neighboring type II cluster (μ_3_ type, binding two Mo in the type II cluster and one Mo from the type II cluster). Thus, only terminal oxo ligands (Mo=O) and a one of μ-O^i^ ligand of type II cluster do not participate in the binding of clusters to each other. Based on the coordination of the ligands, the structural formula of the anions can be described as [(Mo^V^_2_)_6_O_12_(μ-O)_4_(μ_3_-O)_12_(μ-L)_8_]^4−^. The Mo-Mo, Mo-O and Mo-N distances in the compounds are in good agreement with literature data for similar complexes (Appendix A).

Compound **1** crystallizes in trigonal space group *R* 3c (Z = 18) and contains the [Mo_12_O_28_(μ-pz)_8_]^4−^ anion, four ammonium cations, 4.5 solvate water molecules and 1.5 solvate pyrazole molecules per formula unit. One pyrazole molecule has a position occupancy of 0.5, while the solvate water molecules are disordered over several positions or have incomplete position occupancy. Compound packing occurs through networks of hydrogen bonds (Appendix A): between ammonium and oxo-ligands (N···O-Mo distances of 2.82–2.88 Å), between ammonium and water molecules (N···O^water^ distances of 2.70–2.73 Å) and between ammonium and solvate pyrazole molecules (N···N^2-pz^ distance of 2.95 Å).

Structure of **2** was modelled with the entire solvation electron density described by water molecules, since the 1,2,4-triazole molecules were not found directly from the difference electron density maps (see the Section 3.4 for more details). Compound **2** crystallizes in cubic space group *I*
$\bar{4}$3d (Z = 12) and contains the [Mo_12_O_28_(μ-1,2,4-trz)_8_]^4−^ anion, four ammonium cations and ten solvate water molecules per formula unit. Solvate molecules are usually disordered or have incomplete position occupancy. The packing of the compounds is realized by a network of hydrogen bonds: (i) each 1,2,4-triazolate ligand interacts with ammonium cations (N^4-trz^···N distances of 2.87–3.01 Å), which also interact with oxo-ligands of neighboring clusters (N···O-Mo distances of 2.88–3.00 Å); (ii) interactions between oxo-ligands and solvate water molecules (Mo-O···O^water^ distances of 2.90 Å) (Appendix A).

Compound **3** crystallizes in monoclinic space group *P* 2_1_/n (Z = 4) and contains the [Mo_12_O_28_(μ-1,2,3-trz)_8_]^4−^ anion, four ammonium cations and four solvate water molecules per formula unit. One of the ammoniums is disordered at two positions with an occupancy of 0.5 each, and the water molecules are disordered at multiple positions or have incomplete occupancy of the positions. POMC packaging is achieved by a complex network of hydrogen bonds between ammonium cations, solvated water molecules, oxo ligands, and 1,2,3-triazolate ligands (N···O-Mo, Mo-O···O^water^, N^3-trz^···O^water^ distances of 2.71–2.93, 2.79–2.96, and 2.92–3.07 Å, correspondingly).

### 2.3. NMR Spectroscopy and Mass Spectrometry

The complexes were characterized in aqueous solution by NMR spectroscopy (Figure 3a and Appendix A). For example, the ^1^H NMR spectrum (Figure 3a) of compound **1** contains five signals related to coordinated pyrazolate ligands (signals 3–5) and two signals of the pyrazole solvate (signals 1 and 2). When pyrazole is coordinated to the complex, all the signals of the ligand protons are downfield shifted, and the signals of the protons at the 3 and 5 positions of the ring (labelled 1 for pyrazole and 3 and 5 for the pyrazolate ligand in Figure 3a) have a larger shift (Δ~0.6–0.9 ppm for signals 3 or 5 and Δ~0.2 ppm for signals 4) because they are closest to the coordination center.

Despite the apparent symmetry of the compounds and the equivalence of the ligands, the signals of the protons in the 3 and 5 positions of the ring are divided into a group of four signals (integral intensity ratio 1:1:1:1). In addition, according to multinuclear NMR (Appendix A), signals at 8.30 ppm and 8.60 ppm refer to one pyrazolate ligand, signals at 8.39 and 8.63 to another, and a signal at 6.60 ppm refers to protons in the 4th position of all pz ligands (the integral intensity is twice as high as the signals described above). This difference in the position of the signals can be explained by the different environments of the binuclear clusters. Thus, as mentioned above, there are two types of clusters in the structure: with four and two coordinated pyrazolate ligands. On the other hand, these two types of clusters can be connected to each other in different ways: (i) by two oxo ligands and one pyrazolate ligand (Figure 3a, top left) and (ii) by one oxo ligand and one pyrazolate ligand (Figure 3a, top right). All these factors can influence the non-equivalence of the ligands and the different shifts of the proton signals in the 3 and 5 positions of the ring to a downfield. Using 2D NMR correlation spectroscopy, it is not possible to assign the NMR signals of pyrazolate ligands to the structure. However, we assume that the largest downfield shift will be observed for the proton signals labelled 3, since the pyrazolate ligand is coordinated on this side to the binuclear cluster in which each molybdenum atom has five oxygen ligands in the coordination sphere (in addition to the pyrazolate ligand), as opposed to four for another type of binuclear cluster, which will contribute to a stronger redistribution of electron density from the ligand to the metal. The HR-ESI-MS spectrum of an aqueous solution of compound **1** contains several anionic forms, such as {HMo_12_O_28_(pz)_8_}^3−^ (*m*/*z* = 712.3234) (Figure 3b) and fragment forms with fewer pz-ligands (Appendix A), formed during ionization of the solution, confirming the composition of POMC.

A similar situation is observed for compound **2** (Figure 4a and Appendix A). Upon coordination of a 1,2,4-triazole ligand, a shift of the signals towards a weak field (Δ∽0.6–0.9 ppm) is observed. The proton signals at the 3 and 5 positions of the triazole ring now represent a group of three signals at 8.92, 9.19 and 9.27 ppm with an integrated intensity ratio of 2:1:1. The signals at 9.19 and 9.27 ppm refer to different triazoles, while the signal at 8.92 ppm refers to the protons of both triazoles. In this case, we assumed that the largest signal shift would be observed for protons labeled 2, due to the same fact as was mentioned for POMC with pyrazole. The HR-ESI-MS spectra also contain the expected anionic forms of POMC, including {HMo_12_O_28_(1,2,4-trz)_8_}^3−^ (Figure 4b) and others with lower amounts of trz-ligands (Appendix A).

In the case of compound **3**, we expected the same results. However, the ^1^H NMR spectrum of the compound cannot be analyzed (Appendix A) because it contains many signals indicating a mixture of products. Nevertheless, the mass spectrum of an aqueous solution of compound **3** contains the necessary anionic forms (for example {HMo_12_O_28_(1,2,3-trz)_8_}^3−^, Figure 5 and Appendix A), confirming the preservation of the structure of the complex in solution. This difference in these analyses indicates the presence in solution of forms with the same composition but with a different structure. In fact, unlike 1,2,4-triazole, in the structure of 1,2,3-triazole, there are three nitrogen atoms side by side, which formally creates two possible ways of ligand coordination. In this case, different coordination will affect the position of the third nitrogen atom in the structure: the atom will be outside the POMC or facing the wall of the “barrel”. Since there are eight such ligands in the structure, their relative arrangement will produce a large number of linkage isomers. Due to the fact that the resulting complex is a mixture of isomers and also has a significantly lower solubility in water, no further studies were performed on the compound.

### 2.4. Stability in Water Solution

Since the compounds are soluble in water, an important factor is their stability in aqueous solutions at different pH. Stability was studied by changes in electronic absorption spectra over two days (C~1 mM, diluted 60 times to record spectra) (Figure 6). Compounds **1** and **2** showed high stability in distilled water (pH~6). In an alkaline solution (NaOH, pH~12), there is a preservation of the spectral profiles and a slight change in the absorption intensity for compound **2** and a decrease in the absorption intensity (~10%) for compound **1**, possibly due to the partial precipitation of the complex from the solution. The most significant changes occur in the acid solution (HCl, pH~2). After two days, in the case of compound **1**, a change in the color of the solution from orange-red to blue is observed, which is also confirmed in the UV–vis spectra by the appearance of absorption at 500–1000 nm, indicating the formation of “molybdenum blues” and the complete destruction not only of the compound but also of the binuclear clusters. Protonation of the pyrazolate ligands, which occurs in acidic solutions, results in the dissociation of binuclear clusters. Compound **2** shows a different behavior—the complex does not dissolve in acidic solution, and when an aqueous solution of **2** is added to the acid, it immediately precipitates. Unlike the pyrazole, the triazole structure has an additional free nitrogen atom that can be protonated. In this case, protonation of the triazolate ligands and formation of a neutral form [Mo_12_O_28_(μ-1,2,4-trz)_4_(μ-1,2,4-trzH)_4_] with low solubility apparently occurs. We assume that the process stops exactly at the neutral form, since a further decrease in pH does not lead to dissolution of the complex; and according to energy-dispersive X-ray spectroscopy (EDS) data, there is no Cl in the compound. The complex can be dissolved back into water by adding any alkali solution, which is confirmed by the preservation of the ^1^H NMR spectra (Appendix A). Typically, most POMs are not stable in weakly acidic or alkaline aqueous solutions [26,27], while the compounds studied show the opposite situation, which is associated with both the structure based on binuclear clusters and their linkages through bridging pyrazolate or triazolate ligands.

### 2.5. Biological Properties

As was mentioned in the introduction, there are prospects for the use of POMs in the fields of biology and medicine [13,14]. More recent work is also devoted to the study of the biological properties of binuclear molybdenum cluster complexes with organic ligands [28,29]. Often, both POMs and clusters are considered to be effective antibacterial, anticancer or antiviral agents [13,15,16,30,31,32,33]. However, the POMCs previously described in the introduction were usually insoluble in water and therefore their biological properties were not studied. As described above, the compounds obtained are stable in aqueous solutions. In addition, the compounds are also stable in the culture medium (Appendix A), which prompted us to make a first evaluation of their cytotoxicity. The cytotoxic effect of compounds **1** and **2** on normal (MRC-5, lung tissue fibroblasts of a human embryo) and cancer (Hep-2, human larynx carcinoma cell line) cell cultures was studied using the MTT assay (Figure 7). According to the data obtained, in the concentration range studied (up to 2 mM), POMCs do not lead to cell death, with the exception of compound **2** on MRC-5 cells, where the half-maximal cell inhibition concentration (IC_50_) was reached and is equal to 1.81 ± 0.04 mM. Unfortunately, there are no studies in the literature on compounds with similar structures/compositions. The biological properties of binuclear clusters have been studied using various other cell lines. However, in these cases, the IC_50_ values were varied up to 100 μM or were not reached due to low concertation studied [28,29]. In our case, compounds **1** and **2** are significantly less toxic than previously studied binuclear molybdenum clusters to both cancer and normal cells. This observation may be due to the high stability of the compounds, which does not lead to the formation of more toxic products. Cellular penetration of compounds **1** and **2** was assessed after 30 min, 2, 6, 24 and 48 h of incubation by determination of the molybdenum content in the cells using inductively coupled plasma atomic emission spectroscopy (ICP-AES) (Appendix A). According to the data obtained, the molybdenum content in the cells was very low in all cases (the highest value is 0.83 µg_Mo_ per 5·10^5^ cells vs. initial ~2350 µg_Mo_/5·10^5^ cells), indicating the absence of cell penetration.

Since no significant anticancer effect was found, the antiviral properties of the compounds were also investigated. The strain of influenza A virus, subtype H5N1, was used as a model virus. The panzootic caused highly pathogenic avian influenza (HPAI) A(H5) viruses have occurred in multiple waves since 1996. Since 2013, its viruses have emerged to cause panzootic waves of unprecedented magnitude among avian species accompanied by severe losses to the poultry industry worldwide [34]. To date, these viruses have caused only sporadic human infections and are unable to transmit efficiently among humans. However, due to the lack of human population immunity and the ongoing evolution of the virus, there is a continuing risk that clade 2.3.4.4 A(H5) viruses could cause an influenza pandemic if the ability to transmit efficiently among humans was gained [35].

Antiviral activity was assessed by changes in the viability of MDCK cells (Madin–Darby canine kidney cells) in the presence of virus and POMC (Figure 8), i.e., indirect determination of antiviral effect. The MDCK cell line is widely used in virological research for the isolation and production of respiratory viruses, including human viruses, due to the presence of glycan receptors on the cell surface [36]. Virological studies of various influenza virus strains, including H5N1, are performed on MDCK cells [37]. In the absence of virus, compounds **1** and **2** do not show any significant cytopathic effect on MDCK cells (Figure 8a). When tested in the presence of virus, the antiviral effect of compound **1** was found at concentrations above 62.5 μM, while compound **2** showed no effect or insignificant effect in the concentration range tested (up to 500 μM) (Figure 8b). The calculated EC_50_ (by GraphPad Prism software (version 9.1.1) (nonlinear regression dose–inhibition curve)) for compound **1** in the presence of virus is equal to 82.4 ± 11.4 μM (the 50% virus-inhibitory effective concentration as indirectly measured by MDCK cell viability). For comparison, Cs_2_K_4_Na[SiW_9_Nb_3_O_40_] (POM93), known from the literature to have antiviral activity against Influenza A H1N1, had an EC_50_ of 7.4 ± 1.1 μg/mL (∽2.4 μM, incubation time was 48h) [38], which is an order of magnitude lower than for compound **1**. On the other hand, POM93 exhibits marked cytotoxicity against MDCK cells (IC_50_ = 139.9 μM), while the IC_50_ value for the compounds studied is >500 μM. Nevertheless, POMC are not superior in antiviral properties to known antiviral agents such as zanamivir, oseltamivir and their derivatives whose EC_50_ values are usually lower than 10 μM [39,40]. At present, we do not know the exact mechanism of the antiviral action of POMC. However, we believe that further studies of such compounds may contribute to the development of a new class of antiviral agents.

## 3. Materials and Methods

### 3.1. Chemicals and Materials

(NH_4_)_2_[Mo_2_O_4_(C_2_O_4_)_2_(H_2_O)_2_]·3H_2_O was obtained by the reaction of (NH_4_)_6_Mo_7_O_24_·4H_2_O with N_2_H_4_·2HCl and oxalic acid [41]. All other reactants and solvents were purchased from Fisher (Hampton, NH, USA), Alfa Aesar (Haverhill, MA, USA) and Sigma-Aldrich (St. Louis, MO, USA) and used as received.

### 3.2. Syntheses

#### 3.2.1. (NH_4_)_4_[Mo_12_O_28_(μ-pz)_8_]·1.5pzH·4.5H_2_O (Denoted as 1)

(NH_4_)_2_[Mo_2_O_4_(C_2_O_4_)_2_(H_2_O)_2_]·3H_2_O (200 mg, 358 μmol) and pyrazole (200 mg, 2.938 mmol) were ground and heated in a sealed at ambient conditions glass tube at 200 °C for 2 days. The reaction mixture was slowly cooled to room temperature at a rate of 7.5 °C/h. The sealed tube after the synthesis is under high pressure due to the gas formed by the decomposition of oxalic acid and oxalates. The reaction mixture was washed with diethyl ether (∽30 mL), with cold water (∽2 mL), ethanol (∽10 mL), and dried in air. Yield: 136 mg (95% based on (NH_4_)_2_[Mo_2_O_4_(C_2_O_4_)_2_(H_2_O)_2_]·3H_2_O). Anal. Calcd. for C_28.5_H_55_Mo_12_N_23_O_32.5_: C, 14.3; H, 2.3; N, 13.5. Found: C, 14.6; H, 2.1; N, 13.5. FTIR (KBr, cm^−1^): all expected peaks for the pyrazole ligand were observed (Appendix A). The TGA revealed a weight loss of ∼9.0% from 25 to 210 °C (the calculated weight loss of 4.5 H_2_O and 1.5 pzH is 7.7%) followed by slow decomposition of POMC (Appendix A). ^1^H NMR (500 MHz, DSS) δ 6.42 (t, 0.6H, J = 2.0 Hz, H4-pzH), 6.60 (t, 1H, J = 2.0 Hz, H4-pz^2^), 6.61 (t, 1H, J = 2.0 Hz, H4-pz^1^), 7.70 (d, 1.2H, H3-, H5-pzH), 8.30 (d, 1H, H3-, H5-pz^2^), 8.39 (d, 1H, H3-, H5-pz^1^), 8.60 (d, 1H, H3-, H5-pz^2^), 8.63 (d, 1H, H3-, H5-pz^1^). HR-ESI-MS (–) water: 1068.9810 ({H_2_Mo_12_O_28_(pz)_8_}^2−^), 1034.9680 ({HMo_12_O_28_(pz)_7_}^2−^), 1000.9547 ({Mo_12_O_28_(pz)_6_}^2−^), 712.3234 ({HMo_12_O_28_(pz)_8_}^3−^), 689.6480 ({Mo_12_O_28_(pz)_7_}^3−^) (Appendix A). UV–vis (H_2_O): λ_max_, nm (ε, M^−1^ cm^−1^): 300 ((2.8 ± 0.2) × 10^4^) The single crystals of compound **1** suitable for X-ray structural analyses were separated manually from the melts. Diffraction pattern of compound **1** is in good agreement with the theoretical one from SCXRD data (Appendix A).

#### 3.2.2. (NH_4_)_4_[Mo_12_O_28_(μ-1,2,4-trz)_8_]·1.5(1,2,4-trzH)·3H_2_O (Denoted as 2)

Same procedure as in **1**, but 1,2,4-triazole (200 mg, 2.896 mmol) was used instead of pyrazole. Yield: 130 mg (91% based on (NH_4_)_2_[Mo_2_O_4_(C_2_O_4_)_2_(H_2_O)_2_]·3H_2_O). Anal. Calcd. for C_19_H_42.5_Mo_12_N_32.5_O_31_: C, 9.6; H, 1.8; N, 19.2. Found: C, 9.8; H, 1.9; N, 18.9. FTIR (KBr, cm^−1^): all expected peaks for the 1,2,4-triazole ligand were observed (Appendix A). The TGA revealed a weight loss of ∼2.0% from 25 to 150 °C (the calculated weight loss of 3 H_2_O is 2.3%) followed by removing of solvate 1,2,4-trzH and slow decomposition of POMC (Appendix A). ^1^H NMR (500 MHz, DSS) δ 8.32 (s, 0.48H, TrzH), 8.918 (s, 1H, H3-, H5-1,2,4-trz^1^), 8.921 (s,1H, H3-, H5-1,2,4-trz^2^), 9.19 (s, H3-, H5-1,2,4-trz^2^), 9.27 (s, 1H, H3-, H5-1,2,4-trz^1^). ^13^C NMR (126 MHz, DSS) δ 148.87 (d, ^1^J_CH_ = 211.6 Hz, TrzH), 154.24 (dd, ^1^J_CH_ = 210.1 Hz, ^3^J_CH_ = 9.1 Hz, 1,2,4-trz^2^), 154.83 (dd, ^1^J_CH_ = 210.0 Hz, ^3^J_CH_ = 9.4 Hz, 1,2,4-trz^2^), 156.75 (dd, ^1^J_CH_ = 212.2 Hz, ^3^J_CH_ = 9.4 Hz, 1,2,4-trz^1^), 157.37 (dd, ^1^J_CH_ = 212.2 Hz, ^3^J_CH_ = 9.4 Hz, 1,2,4-trz^1^). ^15^N NMR (51 MHz, HCONH_2_) δ 252 (1,2,4-trz^2^), 259 (1,2,4-trz^1^), 260 (1,2,4-trz^2^). HR-ESI-MS (–) water: 1072.9702 ({H_2_Mo_12_O_28_(1,2,4-trz)_8_}^2−^), 1049.4486 ({NaMo_12_O_28_(1,2,4-trz)_7_}^2−^), 1038.4565 ({HMo_12_O_28_(1,2,4-trz)_7_}^2−^), 1003.9381 ({Mo_12_O_28_(1,2,4-trz)_6_}^2−^), 714.9762 ({HMo_12_O_28_(1,2,4-trz)_8_}^3−^), 691.9689 ({Mo_12_O_28_(1,2,4-trz)_7_}^3−^) (Appendix A). UV–vis (H_2_O): λ_max_, nm (ε, M^−1^ cm^−1^): 310 ((2.75 ± 0.02) × 10^4^) The single crystals of compound **2** suitable for X-ray structural analyses were separated manually from the melts. Diffraction pattern of compound **2** is in good agreement with the theoretical one from SCXRD data (Appendix A).

#### 3.2.3. (NH_4_)_4_[Mo_12_O_28_(μ-1,2,3-trz)_8_]·4H_2_O (Denoted as 3)

Same procedure as in **1**, but 1,2,3-triazole (168 μL, 200 mg, 2.896 mmol) was used instead of pyrazole. Yield: 126 mg (92% based on (NH_4_)_2_[Mo_2_O_4_(C_2_O_4_)_2_(H_2_O)_2_]·3H_2_O). Anal. Calcd. for C_16_H_40_Mo_12_N_28_O_32_: C, 8.4; H, 1.8; N, 17.1. Found: C, 8.6; H, 1.8; N, 17.1. FTIR (KBr, cm^−1^): all expected peaks for the 1,2,3-triazole ligand were observed (Appendix A). The TGA revealed a weight loss of ∼3.7% from 25 to 130 °C (the calculated weight loss of 4 H_2_O is 3.1%) and stability of complex up to 250 °C (Appendix A). HR-ESI-MS (–) water: 1049.4486 ({NaMo_12_O_28_(1,2,3-trz)_7_}^2−^), 1038.4519 ({HMo_12_O_28_(1,2,3-trz)_7_}^2−^), 1003.9381 ({Mo_12_O_28_(1,2,3-trz)_6_}^2−^), 978.4289 ({Mo_12_O_28_(1,2,3-trz)_5_(OH)}^2−^), 714.9763 ({HMo_12_O_28_(1,2,3-trz)_8_}^3−^), 691.9688 ({Mo_12_O_28_(1,2,3-trz)_7_}^3−^), 651.9506 ({Mo_12_O_28_(1,2,3-trz)_5_OH}^3−^) (Appendix A). UV–vis (H_2_O): λ_max_, nm (ε, M^−1^ cm^−1^): 310 ((2.87 ± 0.06) × 10^4^) The single crystals of compound **3** suitable for X-ray structural analyses were separated manually from the melts. Diffraction pattern of compound **3** is in good agreement with the theoretical one from SCXRD data (Appendix A).

### 3.3. Physical Methods

Elemental analyses were obtained using a vario MICRO cube analyzer and EuroVector EA3000. Energy-dispersive X-ray spectroscopy (EDS) was performed on a Hitachi TM3000 TableTop SEM (Hitachi High-Technologies Corporation, Tokyo, Japan) with Bruker QUANTAX 70 EDS equipment. FTIR spectra were recorded on a Bruker Vertex 80 as KBr disks. The thermal properties (TGA) were studied on a Thermo Microbalance TG 209 F1 Iris (NETZSCH) from 25 to 800 °C at the heating rate of 10 °C·min^−1^ in He flow (30 mL·min^−1^). Powder X-ray diffraction patterns were collected on a Philips PW 1820/1710 diffractometer (CuK_α_ radiation, graphite monochromator and Si as an external reference). The absorption spectra (UV–vis) were recorded on an Agilent Cary 60 UV/Vis spectrophotometer.

The high-resolution electrospray mass spectrometric (HR-ESI-MS) detection was performed at the Center of Collective Use «Mass spectrometric investigations» SB RAS in negative mode within 500–3000 *m*/*z* range on an electrospray ionization quadrupole time-of-flight (ESI-Q-TOF) high-resolution mass spectrometer Maxis 4G (Bruker Daltonics, Bremen, Germany). The 1D and 2D NMR spectra of sample were obtained from D_2_O solution at room temperature on a Bruker Avance III 500 FT-spectrometer with working frequencies 499.93, 125.71 and 50.66 MHz for ^1^H, ^13^C and ^15^N, respectively. The ^1^H and ^13^C NMR chemical shifts are reported in ppm of the δ scale and referred to the signal of the methyl group of the internal standard of sodium 3-(trimethylsilyl)propane-1-sulfonate (DSS, δ = 0.00 ppm. The ^15^N NMR chemical shifts are referred to external standard of formamide (δ (^15^N) = 112.5 ppm).

### 3.4. Crystallography

Single-crystal X-ray diffraction data for **1**, **2** and **3** at 150 K on a Bruker D8 Venture diffractometer fitted with graphite monochromatized MoKα radiation (λ = 0.71073 Å) were obtained. Absorption corrections were made empirically using the SADABS program [42]. The structures were solved by the direct method and further refined by the full-matrix least-squares method using the SHELXTL (Version 06.12) program package [42]. All non-hydrogen atoms were refined anisotropically. The structure of **2** was modelled with the entire solvation electron density described by water molecules, since the 1,2,4-triazole molecules were not found directly from the difference electron density maps. Elemental analysis indicates the presence of 1.5 molecules of 1,2,4-triazole and 3 molecules of water per formula unit. Analysis of a model, in which an electron density was not described by any solvent, using PLATON SQUEESE methods gives the free volume for solvates at 347 Å^3^ and 101 electrons per formula unit. The electron density for solvate molecules found by chemical analysis and the unlocalized H atoms of ammonium cations is 94 electrons per formula unit, which is in good agreement with the PLATON SQUEESE results. Appendix A summarizes crystallographic data, while CCDC 2299185-2299187 contains the supplementary crystallographic data for this paper. These data can be obtained free of charge from the Cambridge Crystallographic Data Centre via www.ccdc.cam.ac.uk/data_request/cif (accessed on 4 October 2023).

### 3.5. Cell Culture

MDCK (Madin–Darby canine kidney, ATCC CCL34), MRC-5 (lung tissue fibroblasts of a human embryo, collection code 112) and Hep-2 (human larynx carcinoma) cell lines were obtained from the Federal Budgetary Research Institution State Research Center of Virology and Biotechnology “Vector” Rospotrebnadzor cell culture collection, Russian Federation and cultured in Eagle’s Minimum Essential Medium (EMEM) and Dulbecco’s Modified Eagle’s Medium (DMEM) in relation to 1:1 supplemented with a 10% fetal bovine serum under a humidified atmosphere (5% CO_2_ and 95% air) at 37 °C.

### 3.6. The MTT Test

The effect of compounds **1**, **2** and **3** on the cells metabolic activity was determined using the 3-[4,5-dimethylthiazol-2-yl]-2,5-diphenyltetrazolium bromide (MTT) colorimetric assay. The Hep-2 and MRC-5 cells were seeded into 96-well plates at 7 × 10^3^ cells/well in a medium-containing compound with concentrations from 2 to 0.01 mM and then incubated for 72 h under 5% CO_2_ atmosphere. After that, 10 μL of the MTT solution (final concentration of MTT is 0.5 mg·mL^−1^) was added to each well, and the plates were incubated for a further 4 h. The formazan produced was then dissolved in DMSO (100 μL/well). The optical density of the solutions was measured with a Mindray Microplate Reader MR-96A (Mindray, Shenzhen, China) at a wavelength of 492 nm. The percentage of cell viability was calculated by the formula: (optical density of the solution in the wells after incubation with the compounds/optical density of the control solution) × 100%. Wells with cells incubated in a pure nutrient medium were used as a control.

### 3.7. Inductively Coupled Plasma Atomic Emission Spectroscopy (ICP-AES)

The cells were seeded into 6-well plates at concentration of 5 × 10^5^ cells/well. The next day, solutions of **1** and **2** were added to the plates to give final concentrations of 1.0 mM and incubated with cells for 30 min, 2, 6, 24 and 48 h. Then, cells were washed three times with a sterile phosphate buffer. The cells were then removed from the wells with trypsin and centrifuged at 1000 rpm for 5 min. After centrifugation, the number of cells in each sample was calculated. The resulting cell suspension was brought to a volume of 300 µL in a sterile buffer so that each sample contained 0.5 million cells.

Molybdenum content in cells after incubation times was determined on a high-resolution spectrometer iCAP-6500 (Thermo Scientific, Waltham, MA, USA) with a cyclone-type spray chamber and “SeaSpray” nebulizer. The spectra were obtained by axial plasma viewing. Standard operating conditions of the ICP-AES system were following: power = 1150 W, injector inner diameter = 3 mm, carrier argon flow = 0.7 L min^−1^, accessorial argon flow = 0.5 L min^−1^, cooling argon flow = 12 L min^−1^, number of parallel measurements = 3, and integration time = 5 s.

### 3.8. The Antiviral Activity Assay

To evaluate the antiviral effect, a strain of influenza virus type A, subtype H5N1 (strain A/chicken/Chelyabinsk region/2KL/Russia/2021) was used as a model virus. To evaluate the antiviral effect of samples **1** and **2**, compounds were added in concentrations from 500 µM to 1.95 µM to a subconfluent monolayer of MDCK cells in a 96-well culture plate (TPP, Trasadingen, Switzerland), washed twice with Hanks’ solution (Biolot, Saint-Petersburg, Russia). To study the antiviral effect of the samples against influenza A virus (H5N1), 200 μL of virus-containing liquid lg2 TCID_50_/mL was added to the test samples in the wells of the plate. As a viral control, 1 mL of virus-containing liquid lg3 TCID/mL was added to the subconfluent cell monolayer. The virus-containing liquid test samples were incubated for 72 h in a CO_2_ incubator under conditions of 37 °C, 5% CO_2_. The CPE (cytopathic effect) of influenza virus on cells was evaluated under an inverted microscope (Mikromed, Moscow, Russia) for 3 days, and the MTT test was performed (see Section 3.6 The MTT test). Six independent measurements of the optical density of the solution were made for each test sample and control during the MTT assay. Virus activity was calculated by the Spearman–Kerber method [43,44].

## 4. Conclusions

In conclusion, in this work, new POMs have been obtained in high yields, leading to a new class of compounds—polyoxometal clusters (POMCs)—which combine the structural features of POMs and cluster complexes. The compounds (NH_4_)_4_[Mo_12_O_28_(μ-L)_8_], obtained by ampoule synthesis from a binuclear cluster, are soluble and stable in aqueous solutions (at neutral and alkaline pH). The compounds were characterized in aqueous solution using NMR spectroscopy (presence of signals from coordinated ligands) and mass spectrometry (presence of anionic forms of the required composition). The stability of the compounds in water allowed us to perform first studies on the cytotoxicity of the compounds on normal (MRC-5) and cancerous (Hep-2) cell cultures. In the concentration range studied (up to 2 mM), only one compound with 1,2,4-triazole reached an IC_50_ value of 1.81 ± 0.04 mM on MRC-5 cells. Despite the fact that the complexes show no significant toxic effect on cancer cell cultures, POMC with pyrazole shows antiviral activity at concentrations above 62.5 μM. We believe that the results obtained will serve as a basis for further development of this class of compounds and for more detailed studies of their biological properties. Further work, including with functional organic compounds (containing pyrazole and triazole fragments), whose investigation has already shown potential for use as antiviral agents, may then lead to a synergistic effect of POMCs and ligands, resulting in the creation of new agents for biology and medicine.

## Figures and Tables

**Figure 1 molecules-28-08079-f001:**
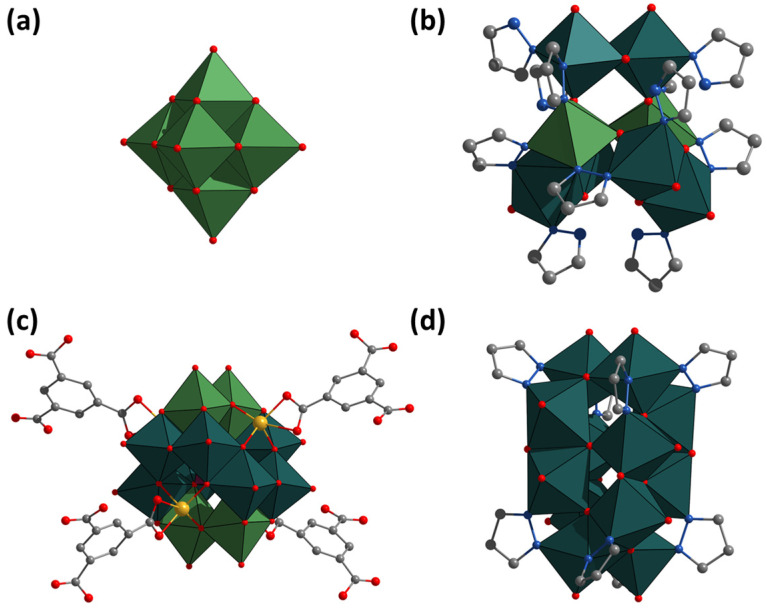
Structure of known in the literature and obtained in this work compounds: (**a**) [Mo^VI^_6_O_19_]^2−^, (**b**) [Mo^V^_6_Mo^VI^_2_(μ-pz)_6_O_18_(pzH)_6_], (**c**) [[PMo^V^_8_Mo^VI^_4_O_36_(OH)_4_Zn_4_][C_6_H_3_(COO)_3_]_4/3_]^3−^, and (**d**) [Mo_12_O_28_(μ-pz)_8_]^4−^. Color code: O—red, C—gray, N—blue, Zn—yellow, Mo^V^_2_O_6_N_4_ and Mo^V^O_8_N_2_—blue-green edge-connected octahedrons, MoO_6_—green octahedrons, and PO_4_—violet octahedron.

**Figure 2 molecules-28-08079-f002:**
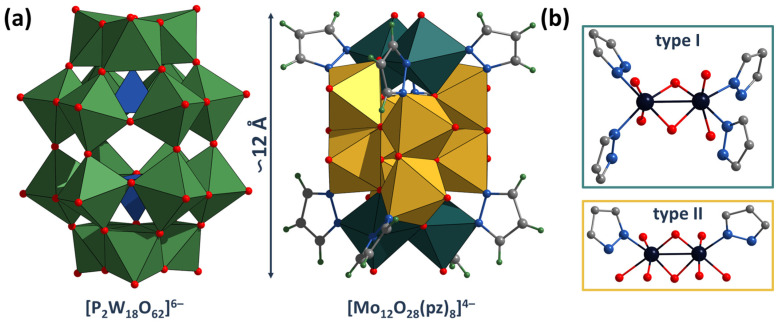
(**a**) Size and structure comparison of POMC [Mo_12_O_28_L_8_]^4−^ with Dawson-type POM [P_2_W_18_O_62_]^6−^. (**b**) Different ligand environment of binuclear clusters in POMC. Color code: Mo—dark blue, O—red, C—gray, N—blue, H—light green, Mo^V^_2_O_6_N_4_—yellow edge-connected octahedrons, Mo^V^O_8_N_2_—blue-green edge-connected octahedrons, WO_6_—green octahedron, and PO_4_—blue octahedron.

**Figure 3 molecules-28-08079-f003:**
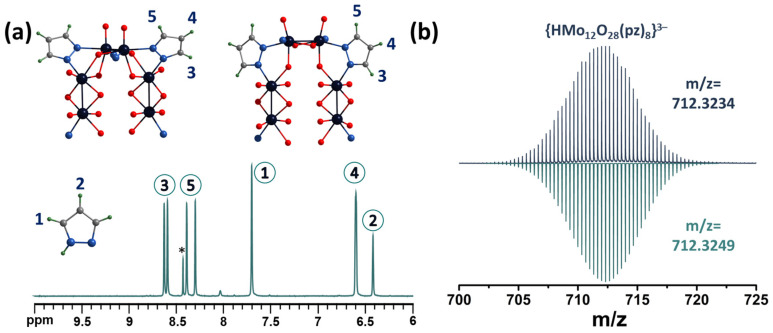
(**a**) Fragment of ^1^H NMR spectrum of compound **1** in D_2_O. (**b**) Fragment of mass spectrum of compound **1** in H_2_O indicating form {HMo_12_O_28_(μ-pz)_8_}^3−^ (top spectra—experimental data, bottom spectra—calculated). The asterisk on the NMR spectra indicates the signal of an impurity formed during the synthesis by the decomposition of oxalic acid, which is probably formic acid.

**Figure 4 molecules-28-08079-f004:**
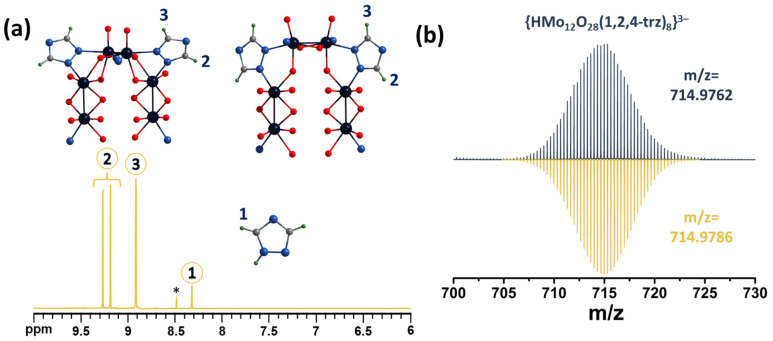
(**a**) Fragment of ^1^H NMR spectrum of compound **2** in D_2_O. (**b**) Fragment of mass spectrum of compound **2** in H_2_O indicating form {HMo_12_O_28_(μ-1,2,4-trz)_8_}^3−^ (top spectra—experimental data, bottom spectra—calculated). The asterisk on the NMR spectra indicates the signal of an impurity formed during the synthesis by the decomposition of oxalic acid, which is probably formic acid.

**Figure 5 molecules-28-08079-f005:**
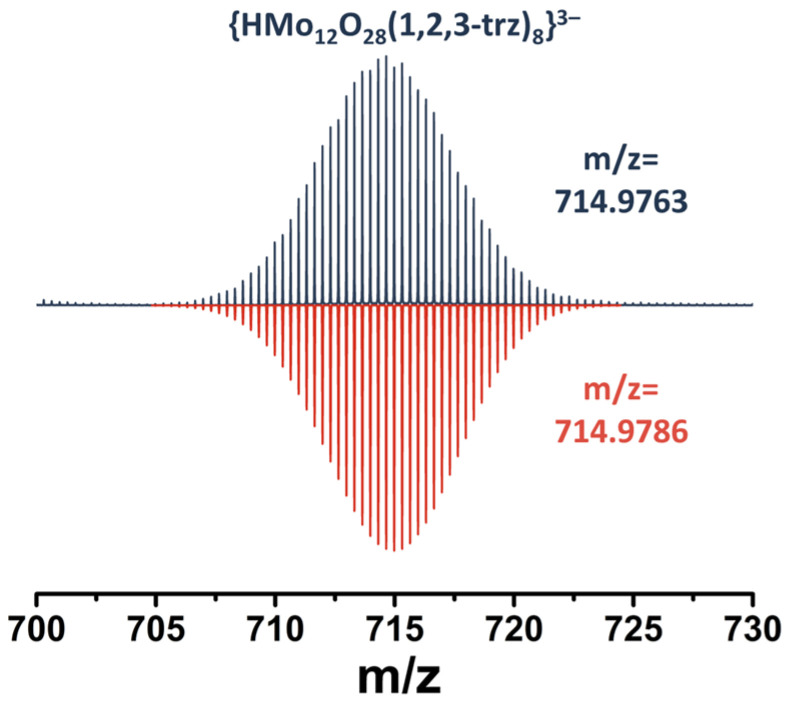
Fragment of mass spectrum of **3** in H_2_O indicating form {HMo_12_O_28_(μ-1,2,3-trz)_8_}^3−^ (top spectra—experimental data, bottom spectra—calculated).

**Figure 6 molecules-28-08079-f006:**
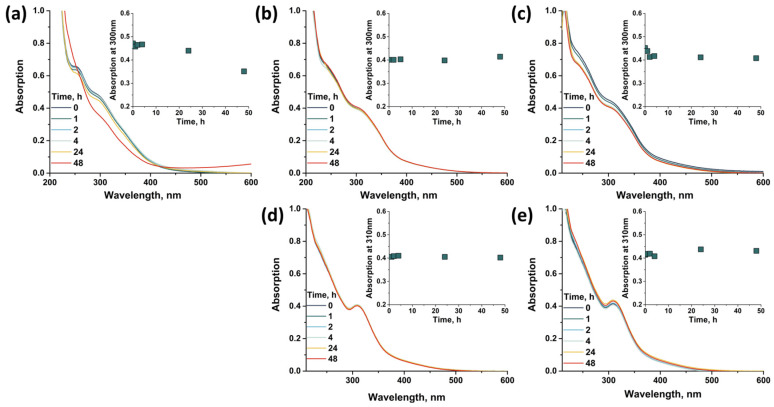
UV–vis spectra of compounds **1** (**a**–**c**) and **2** (**d**,**e**) in water in time at pH = 2 (**a**), 6 (**b**,**d**) and 12 (**c**,**e**). The insets show the change in the absorption band at 300 and 310 nm over time for compound **1** and **2**, respectively.

**Figure 7 molecules-28-08079-f007:**
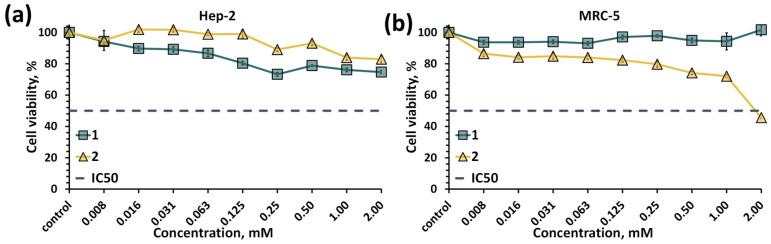
Viability of Hep-2 (**a**) and MRC-5 (**b**) cells treated with compound **1** or **2** determined by the MTT test.

**Figure 8 molecules-28-08079-f008:**
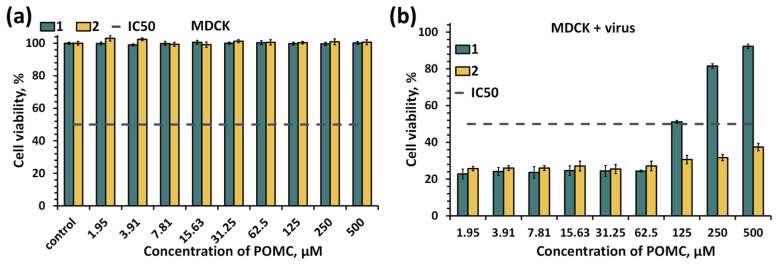
(**a**) Viability of MDCK cells treated with compound **1** or **2** in absence (**a**) or presence (**b**) of influenza A virus, subtype H5N1 determined by the MTT test.

**Table 1 molecules-28-08079-t001:** Compounds based on molybdenum oxo-clusters known from the literature.

Compound	Starting Compound	Yield	Reference
[Mo^V^_6_Mo^VI^_2_(μ-pz)_6_O_18_(pzH)_6_] (pzH = pyrazole) (Figure 1b)	Molybdenum blue	29%	[18]
[Mo^V^_8_O_16_(OCH_3_)_8_L_4_] (L = 4-methylpyridine (4-Me-Py), pyridine (Py))	(PyH)_2_[Mo^V^OCl_5_]	few crystals	[19,20]
[Mo_8_^V^Mo_2_^VI^O_26_L_8_] (L = 3,5-lutidine, 3-methylpyridine (3-Me-Py))	(PyH)_2_[Mo^V^OCl_5_]/(NH_4_)_2_[Mo^V^OCl_5_(H_2_O)]	9–30%	[21]
[Mo^V^_12_O_28_(OCH_3_)_2_Cl_2_(3-Me-Py)_8_]	(PyH)_2_[Mo^V^OCl_5_]	28%	[21]
[Mo^V^_12_O_28_(OC_2_H_5_)_4_(4-Me-Py)_8_]	(PyH)[Mo^V^OBr_4_(H_2_O)]	75%	[22]
[Mo^V^_8_Mo^VI^_2_O_26_(4-*i*Pr-Py)_8_]	(PyH)_2_[Mo^V^OCl_5_]	few crystals	[23]
[Mo^V^_8_Mo^VI^_2_O_26_(py)_8_]	K_2_[Mo^IV^_3_O_4_(Hnta)_3_]	20%	[24]
[Mo^IV^_6_Mo^VI^_4_O_24_(py)_8_]	(H_2_bipy)[Mo^IV^_3_O_4_(C_2_O_4_)_3_(H_2_O)_3_]	60%	[24]
[Mo_8_O_20_(μ-1,2,3-trzH)_8_] (1,2,3-trzH = 1,2,3-triazole)	Na_2_Mo^VI^O_4_	90%	[25]

## Data Availability

Crystal structure data can be obtained free of charge from The Cambridge Crystallographic Data Centre via www.ccdc.cam.ac.uk/data_request/cif (accessed on 4 October 2023) or are available on request from the corresponding author.

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
