# Peer review of "Water-Soluble Polyoxometal Clusters of Molybdenum (V) with Pyrazole and Triazole: Synthesis and Study of Cytotoxicity and Antiviral Activity"

_molecules, 2023, doi:10.3390/molecules28248079_

Round 1
Reviewer 1 Report
Comments and Suggestions for Authors
The manuscript titled "Water-soluble Polyoxometal Clusters of Molybdenum (V) with Pyrazole and Triazole for Biomedical Applications" is a well-conducted study that contributes to the field of Polyoxometal Clusters (POMCs) and their potential biomedical applications. The authors have introduced a new subclass of polyoxometal clusters, which are synthesized and characterized methodically. The manuscript is well-structured, and the methodology employed is sound. The findings presented show promise in the biomedical realm, especially concerning antiviral activity against the H5N1 influenza subtype. However, there are some aspects where minor revisions are recommended for clarity, completeness, and further validation of the findings:
1. It would be beneficial if the authors provide more details on the structural characterization of the synthesized POMCs.
2. The antiviral activity against influenza A virus subtype H5N1 is a significant finding. However, the manuscript could be strengthened by including more details on the antiviral assay protocols, and if possible, comparison with other known antiviral agents, for example H9N2.
3. The conclusion mentions the potential for further development of this class of compounds with more complex functional ligands. It would be valuable if the authors could elaborate more on the envisioned functional ligands and the anticipated impact on the biological properties of the POMCs.
4. The introduction could be enhanced by providing a more extensive literature review, especially regarding the previous studies on transition metals oxide and their applications in catalysis and batteries, such as Adv. Mater. (10.1002/adma.202305074); J. Energy Chem. (10.1016/j.jechem.2023.03.033).
Comments on the Quality of English LanguageMinor editing of English language required
Author Response
Referee 1
The manuscript titled "Water-soluble Polyoxometal Clusters of Molybdenum (V) with Pyrazole and Triazole for Biomedical Applications" is a well-conducted study that contributes to the field of Polyoxometal Clusters (POMCs) and their potential biomedical applications. The authors have introduced a new subclass of polyoxometal clusters, which are synthesized and characterized methodically. The manuscript is well-structured, and the methodology employed is sound. The findings presented show promise in the biomedical realm, especially concerning antiviral activity against the H5N1 influenza subtype. However, there are some aspects where minor revisions are recommended for clarity, completeness, and further validation of the findings:
1.It would be beneficial if the authors provide more details on the structural characterization of the synthesized POMCs.
Answer: An additional discussion of the structure of POMCs was added in section 2.2. Crystal Structure.
2.The antiviral activity against influenza A virus subtype H5N1 is a significant finding. However, the manuscript could be strengthened by including more details on the antiviral assay protocols, and if possible, comparison with other known antiviral agents, for example H9N2.
Answer: Thank you for the good suggestion. An additional discussion and a comparison have been added in the section 2.5 Biological properties.
3.The conclusion mentions the potential for further development of this class of compounds with more complex functional ligands. It would be valuable if the authors could elaborate more on the envisioned functional ligands and the anticipated impact on the biological properties of the POMCs.
Answer: We have extended the conclusion part. Ideally, it would be interesting to coordinate organic compounds that already have antiviral properties (see, for example, references https://doi.org/10.1186/1423-0127-17-13 or https://doi.org/10.1080/15257770.2019.1674331). However, achieving this goal will undoubtedly occur sequentially, complicating step by step the structure of the organic ligand.
4.The introduction could be enhanced by providing a more extensive literature review, especially regarding the previous studies on transition metals oxide and their applications in catalysis and batteries, such as Adv. Mater. (10.1002/adma.202305074); J. Energy Chem. (10.1016/j.jechem.2023.03.033).
Answer: Thank you for providing interesting and important works. However, we believe that these works do not correspond to the objects of this work and belong to another class of compounds – perovskites, comparison with which is not entirely appropriate.
Reviewer 2 Report
Comments and Suggestions for Authors
The manuscript describes a promising class of molybdenum (V) polyoxometal clusters. The main advantages of these types of POMCs are their solubility in water, and the main advantages of the work is the high synthetic yield (more than 90%). This paper is well-prepared and clearly describes the observations.
Possible questions for this manuscript could be
1) Fig. 8 (a and b) should be prepared in the same style. For now, Fig.8a is a Line + Symbol type, and Fig.8b is a Column type.
2) TG data can be described more. Each step should be labeled with weight loss %, and with an indication of the observed process.
3) Does free pyrazole or triazole show cytotoxic or antibacterial effects? And can the activity of molybdenum (V) POMCs pyrazolate and triazolate be compared with other water-soluble pyrazolate and triazolate complexes?
Author Response
Referee 2
The manuscript describes a promising class of molybdenum (V) polyoxometal clusters. The main advantages of these types of POMCs are their solubility in water, and the main advantages of the work is the high synthetic yield (more than 90%). This paper is well-prepared and clearly describes the observations.
Possible questions for this manuscript could be
1) Fig. 8 (a and b) should be prepared in the same style. For now, Fig.8a is a Line + Symbol type, and Fig.8b is a Column type.
Answer: The style of the figure was changed according reviewer suggestion.
2) TG data can be described more. Each step should be labeled with weight loss %, and with an indication of the observed process.
Answer: TGA figures were corrected.
3) Does free pyrazole or triazole show cytotoxic or antibacterial effects? And can the activity of molybdenum (V) POMCs pyrazolate and triazolate be compared with other water-soluble pyrazolate and triazolate complexes?
Answer: To the best of our knowledge such organic compounds are not cytotoxic at studied for complexes concentration range. Usually in recent works one can find studies dedicated to different derivatives of pyrazole and triazole, for which there are data on antibacterial, antiviral and etc. effects. The same information is provided for the complexes with pyrazole and triazole derivatives. However, examples of complexes with pyrazole and 1,2,4-triazole are also known: (Htrz)[trans-RuCl4(trz)(DMSO)] (DOI: 10.1021/jm061081y), [M(pzH)4Cl2] (M= Cu2+, Co2+, Ni2+, Zn2+, DOI 10.1007/s11243-021-00466-4), [(pzH)4GaCl2][GaCl4] (DOI 10.1016/S0277-5387(00)83206-4), [Pt2(pzH)4(berenil)2] (DOI 10.1080/14756366.2018.1471687), mer-[RuCl3(DMSO)2(pzH)] (DOI 10.1016/j.jinorgbio.2012.02.022), (H2trz)[cis-RuCl4(N2-trzH)2] (DOI 10.1021/ic034605i). In most cases, these compounds are considered as cytostatic agents (usually IC50 was not higher than 100 μM or higher concentrations were not studied), and studies have been performed in other cell cultures (HT-29, SK-BR-3, A549, EC109, etc.) with which the comparison is not entirely accurate.
Reviewer 3 Report
Comments and Suggestions for Authors
The work is interesting for the perspectives it opens up. As a biologist, I am not qualified to evaluate the purely chemical aspects but as regards cytotoxicity, I have some requests and doubts for the authors.
line 82. although supporting the chemical work, the presentation of the cytological aspects is marginal and incomplete. it is necessary to specify why the authors believe that the new POMCs would enter the cells (they do? how?) and why these should exert the proposed effects, and on which cellular activity (presumably?). References must be added.
line 262. POMC are 'water-insoluble'. In water, do they form aggregates, precipitate, float ...??? And why they are not worth studying biological activity? Microplastics are also insoluble, however.... In addition, cell membranes are made of lipids and cells often internalise 'particles' by endocytosis. Please clarify cell-POMC interaction.
line 264. Similarly, please explain why it is appropriate to evaluate cytotoxicity for a stable compound and why the same is not for unstable compounds, producing degradation products.
line 266. I suggest changing healthy to normal. Ultimately, the idea of ​​'health' is that the cell does what it is designed to do. A cancer cell does, it is 'healthy' in its own way, otherwise, why use them in toxicity testing if already 'suffering'?
figure 7 (and text). Controls are missing. how is viability in cells not exposed to the 1 or 2 compounds? For other comments, see methods.
line 274. Significantly less toxic than what?
line 275. was the stability of the two compounds tested in the culture medium? In addition, do you have any idea about what metabolite(s) can originate once inside the cell? Can Mo dissociate from the complex once inside the medium/cell? Mo has a role in plant and animal cells and may have contributed to the obtained results ?(Mendel and Kruse, BBA, vols 1823, pages 1568-1579)
line 278. please add 'compounds' before 1 or 2.
line 399. explain why these two cell types were chosen. They are of different origin (fibroblasts and epitheloids): why do you expect they give results that can be compared/discussed together?
line 399. what happens to the culture? Do cells proliferate? the number of cells present is an important point to be addressed to read the MTT assay
line 413. as above: it should be specified that cells do not proliferate during the 72 hours assay. It would be also appropriate to check that metabolism of the exposed cells does not vary: in fact, genetic, epigenetic and metabolic factors control the activity of succinate dehydrogenase (and may interfere with MTT results).
line 413, You used MDCK cells. If I am right, these are of canine origin. Why this choice? Why not using the MRC-5 cells? And how do you justify the infection of a canine cell, from the kidneys, with a human virus, attacking the respiratory mucosa? Is it possible that cells reacted in the wrong way to the exposure? Please justify these important points
line 419. CPD stands for....
line 422-426. Move above, in the MTT test.
line 427. add a reference for the method.
Author Response
Referee 3
The work is interesting for the perspectives it opens up. As a biologist, I am not qualified to evaluate the purely chemical aspects but as regards cytotoxicity, I have some requests and doubts for the authors.
line 82. although supporting the chemical work, the presentation of the cytological aspects is marginal and incomplete. it is necessary to specify why the authors believe that the new POMCs would enter the cells (they do? how?) and why these should exert the proposed effects, and on which cellular activity (presumably?). References must be added.
Answer: Since the compounds are completely new, no one knows anything about their cellular penetration, cytotoxicity, etc. Based on the publication of similar compounds and their potential applications as anticancer, antibacterial and antiviral agents, we decided to perform initial biological experiments on the new complexes to determine whether these compounds have any promise in this field. In addition, the penetration of the compounds into cells is not necessary for any biological effects to occur. However, taking this important note into account, we further investigated the cellular penetration of the complexes by ICP-AES analysis on Mo of cells incubated with the complexes. According to the results obtained, the molybdenum content in the cells was very low (although it increased slightly with increasing incubation time), indicating a complete lack of penetration of the complexes into the cells. The discussion was added in the manuscript.
line 262. POMC are 'water-insoluble'. In water, do they form aggregates, precipitate, float ...??? And why they are not worth studying biological activity? Microplastics are also insoluble, however.... In addition, cell membranes are made of lipids and cells often internalise 'particles' by endocytosis. Please clarify cell-POMC interaction.
Answer: Complexes are ionic compounds that exist in solution as solvated ions, which may include some small aggregates. This was not investigated further in this work. In our case, the complexes did not precipitate out of solution during the biological experiments. Detailed study of the interaction of the complexes with cells was not performed and requires complex biological experiments that were not intended for this work.
line 264. Similarly, please explain why it is appropriate to evaluate cytotoxicity for a stable compound and why the same is not for unstable compounds, producing degradation products.
Answer: As mentioned above, we first wanted to make a preliminary study of the cytotoxicity of the compounds under investigation. If something happens to the complexes in the water or cell medium (destruction of the complexes or precipitation), it will be difficult to know what was the reason for any cytotoxic effects. For this reason, the most stable compounds were chosen. However, it is also useful to study unstable compounds because their degradation products may have specific biological effects. However, this was not included in this study.
line 266. I suggest changing healthy to normal. Ultimately, the idea of 'health' is that the cell does what it is designed to do. A cancer cell does, it is 'healthy' in its own way, otherwise, why use them in toxicity testing if already 'suffering'?
Answer: The correction was done.
figure 7 (and text). Controls are missing. how is viability in cells not exposed to the 1 or 2 compounds? For other comments, see methods.
Answer: The controls are not missing. The dots on the left of Fig. 7 correspond to the control (see axis also). Wells containing cells incubated in pure nutrient medium were used as controls. The percentage of cell viability was calculated using the formula: (optical density of the solution in the wells after incubation with the compounds / optical density of the control solution) x 100%. Thus, all cell viability values obtained are relative to the control group.
line 274. Significantly less toxic than what?
Answer: Complexes are less toxic than previously published binuclear molybdenum clusters. The sentence has been added to the text.
line 275. was the stability of the two compounds tested in the culture medium? In addition, do you have any idea about what metabolite(s) can originate once inside the cell? Can Mo dissociate from the complex once inside the medium/cell? Mo has a role in plant and animal cells and may have contributed to the obtained results? (Mendel and Kruse, BBA, vols 1823, pages 1568-1579)
Answer: Thank you for your important comment. An additional experiment to study the stability of the complexes in the culture medium (by changing the absorption spectra) was performed and added to this work (Figure S16). According to the data obtained, the complexes are stable for at least 3 days, which indicates that compounds are not destroyed and Mo is not released into the solution.
line 278. please add 'compounds' before 1 or 2.
Answer: The correction was done.
line 399. explain why these two cell types were chosen. They are of different origin (fibroblasts and epitheloids): why do you expect they give results that can be compared/discussed together?
Answer: These cell cultures were chosen as the first model cells to study biological properties. Both cell cultures are commonly used to study the cytotoxicity of compounds, which allows us to compare the data obtained with previously published data. In this work, we do not directly compare the cytotoxicity of compounds on different cell lines. Undoubtedly, for a more accurate comparison, it is necessary to use cells of similar origin.
line 399. what happens to the culture? Do cells proliferate? the number of cells present is an important point to be addressed to read the MTT assay
line 413. as above: it should be specified that cells do not proliferate during the 72 hours assay. It would be also appropriate to check that metabolism of the exposed cells does not vary: in fact, genetic, epigenetic and metabolic factors control the activity of succinate dehydrogenase (and may interfere with MTT results).
Answer: Undoubtedly, the cells proliferate during the experiment. No cytostatic effect is observed as there is minor cell death (which is also consistent with the relatively low cytotoxicity of the complexes). Cell death is visually observed under a microscope. As mentioned above, all experiments are performed in comparison with control group of cells incubated in pure nutrient medium. Cells incubated with the complexes proliferate at the same rate as the control.
line 413, You used MDCK cells. If I am right, these are of canine origin. Why this choice? Why not using the MRC-5 cells? And how do you justify the infection of a canine cell, from the kidneys, with a human virus, attacking the respiratory mucosa? Is it possible that cells reacted in the wrong way to the exposure? Please justify these important points
Answer: The MDCK cell line is a culture of immortalized canine kidney epithelial cells. It is widely used in virological research for the isolation and production of respiratory viruses, including human viruses, due to the presence of glycan receptors on the cell surface (doi: 10.1371/journal.pone.0075014). Virological studies of various influenza virus strains, including H5N1, are performed on MDCK cells (doi: 10.3390/pathogens10040483). A comparative study by Reina J. et al. showed greater sensitivity of MDCK cell culture (100%) to influenza A virus isolated from clinical samples compared to Vero monkey cell culture (71.4%) and MRC-5 cell culture (57.1%) (doi: 10.1128/jcm.35.7.1900-1901.1997). In WHO recommendations, Madin-Darby canine kidney (MDCK) cells are typically the preferred cell line in which to culture influenza viruses (Manual for the laboratory diagnosis and virological surveillance of influenza. World Health Organization, 2011. https://www.who.int/publications/i/item/manual-for-the-laboratory-diagnosis-and-virological-surveillance-of-influenza). Additional information was added in the manuscript.
line 419. CPD stands for....
Answer: We are sorry for this mistake. It should be CPE (cytopathic effect). The correction has been made.
line 422-426. Move above, in the MTT test.
Answer: This part has been removed because the same procedure is already mentioned in 3.6 MTT test.
line 427. add a reference for the method.
Answer: The references for the method were added.
Reviewer 4 Report
Comments and Suggestions for Authors
The authors described synthesis of water soluble polyoxometal clusters of molybdenum (V) with pyrazole and triazole for biomedical application. In general, the manuscript is a good, valuable work. However, the manuscript should undergo some revision and certain improvements are necessary.
I would recommend the following corrections:
1. Title should be changed according to the real set provided by authors' studies and obtained results. You can specify the antiviral activity.
2. All abbreviations, like POMs, should be explained every first time they appear in text and abstract.
3. Please provide citation: “In contrast to the methods described in the literature for the preparation of various POMCs…”
4. The following articles can be cited:
https://doi.org/10.1002/anie.202201672
https://doi.org/10.1021/ja303084n
https://doi.org/10.1016/j.ccr.2022.214734
DOI: 10.1039/D0BM01015D
5. What kind of calculations did you use to claim: “The calculated ICv50 for 1,… Line 286”, Can you explain?
6. The cell densities used in the cell viability tests are unusually high range for 96 well plates. Can you explain?
7. The MTT concentration should be 0.5 mg /mL
Comments on the Quality of English LanguageMinor editing of English language required
Author Response
Referee 4
The authors described synthesis of water soluble polyoxometal clusters of molybdenum (V) with pyrazole and triazole for biomedical application. In general, the manuscript is a good, valuable work. However, the manuscript should undergo some revision and certain improvements are necessary.
I would recommend the following corrections:
- Title should be changed according to the real set provided by authors' studies and obtained results. You can specify the antiviral activity.
Answer: The title was changed according reviewer suggestion.
- All abbreviations, like POMs, should be explained every first time they appear in text and abstract.
Answer: The corrections were made.
- Please provide citation: “In contrast to the methods described in the literature for the preparation of various POMCs…”
Answer: The references were added.
- The following articles can be cited:
https://doi.org/10.1002/anie.202201672
https://doi.org/10.1021/ja303084n
https://doi.org/10.1016/j.ccr.2022.214734
DOI: 10.1039/D0BM01015D
Answer: Thank you for mentioning these relevant works. References have been added.
- What kind of calculations did you use to claim: “The calculated ICv50 for 1,… Line 286”, Can you explain?
Answer: The value was calculated from the data obtained using GraphPad Prism software (nonlinear regression dose-inhibition curve). Information has been added in the manuscript.
- The cell densities used in the cell viability tests are unusually high range for 96 well plates. Can you explain?
Answer: The MTT assay was performed according to the standard protocol. The cell density is the optimal value for incubation for 72 hours and has been demonstrated in a number of studies investigating such relations (see, for example, DOI 10.29042/2018-3274-3280).
- The MTT concentration should be 0.5 mg /mL
Answer: It is true, that the final concentration of MTT reagent is 0.5 mg/mL because 10 µL of MTT reagent solution (5 mg/mL) is added to 90 µL of cell medium. This sentence has been changed in the text of the manuscript.
Round 2
Reviewer 3 Report
Comments and Suggestions for Authors
Manuscript improved. Can be published in its actual form
Reviewer 4 Report
Comments and Suggestions for Authors
The authors have addressed all my concerns.